# Fortification of Cereal-Based Food with *Lactobacillus rhamnosus* GG and *Bacillus coagulans* GBI-30 and Their Survival During Processing

**DOI:** 10.3390/foods14132250

**Published:** 2025-06-25

**Authors:** Junyan Wang, Peng Wu, Xiao Dong Chen, Aibing Yu, Sushil Dhital

**Affiliations:** 1Department of Chemical Engineering, Monash University, Clayton, VIC 3800, Australia; junyan.wang@monash.edu; 2School of Chemical and Environmental Engineering, College of Chemistry, Chemical Engineering and Material Science, Soochow University, Suzhou 215123, China; p.wu@suda.edu.cn (P.W.); xdchen@mail.suda.edu.cn (X.D.C.)

**Keywords:** *Lactobacillus rhamnosus* GG, *Bacillus coagulans* GBI-30, probiotic, viability, wheat-flour pasta, noodle, novel foods

## Abstract

Functional foods are evolving beyond basic nutrition to address nutrition-related diseases and enhance well-being. While probiotic-fortified products dominate this sector, most remain dairy-based. This study investigated the incorporation of *Lactobacillus rhamnosus* GG and *Bacillus coagulans* GBI-30 into cereal-based pasta and noodles, evaluating bacterial survival during processing and cooking. Extrusion-based pasta production exerted greater stress on *Lactobacillus rhamnosus* GG, whereas *Bacillus coagulans* GBI-30 demonstrated higher thermal resistance. In sheeted noodles, both strains maintained ≥8 log CFU/g viability pre-cooking. After 7 min boiling, *Lactobacillus rhamnosus* GG retained 6.88 log CFU/g and *Bacillus coagulans* GBI-30 5.75 log CFU/g in noodles, meeting the recommended 10^6^–10^7^ CFU/g threshold for probiotic efficacy. Cooking performance analysis revealed lower cooking loss in noodles (2.4–4.04%) versus extruded pasta (10.6–19.05%), indicating superior structural integrity. These results confirm cereal matrices as viable non-dairy carriers for probiotics, with sheeting processes better preserving bacterial viability than extrusion. The findings highlight a practical strategy for developing functional foods that sustain probiotic viability through processing and consumption, potentially enhancing gut microbiota balance. This approach expands probiotic delivery options beyond traditional dairy formats while maintaining therapeutic bacterial concentrations critical for health benefits.

## 1. Introduction

In the modern food industry, foods are developed not only to satisfy basic satiety requirements but also to reduce the risk of nutrition-related diseases and to improve human physiological well-being [1]. This has led to the emergence of a new category of foods, known as “functional foods”. In the 1980s, the term “functional food” was first used in Japan to describe food products fortified with special ingredients for additional health benefits [1,2,3]. Among the various functional food products, those containing probiotics dominate the market due to their positive effects on intestinal health [1]. In addition, probiotic fortification can improve the nutritional quality of food products and extend their shelf life by producing antimicrobial substances [4,5].

The majority of probiotic-fortified foods available on the market are dairy-based fermented products such as cheese and yoghurt [6]. Recently, the demand for non-dairy probiotic foods has emerged due to the allergenic content of dairy products, lactose intolerance, and vegetarian populations [7]. For instance, grains and cereals are nutritious sources of complex carbohydrates, minerals, vitamins, and dietary fibre [8], making them a potential food category to produce innovative probiotic foods. Dietary fibre not only has beneficial physiological effects on the gut, but also can be used as a prebiotic, providing specific non-digestible carbohydrates and promoting the growth of probiotics [9,10,11]. Pasta and noodles are among the most commonly consumed carbohydrate foods worldwide. They can be easily prepared and has a long shelf-life. From these perspectives, they are a promising non-dairy-based food matrix for the delivery of probiotics.

Preserving the stability, activity, and viability of bacterial cells during the probiotification of foods is essential. A minimum of 10^6^ to 10^7^ colony-forming units (CFU)/g of viable bacterial cells is recommended in probiotic food products to maintain their optimum functionality and exert the desirable therapeutic benefits for consumers [12]. It is widely known that the composition of carrier matrices can affect the viability of probiotics [13]. The environment of a food matrix in which probiotic bacteria are embedded is a significant factor in their growth, survival, and functionality [7]. Although probiotics have been extensively used in the dairy industry for decades, their incorporation into pasta or noodle products represents a recent innovation [14]. Recently, *Bacillus coagulans*, which is identified as GRAS (generally recognized as safe) [15], has attracted significant interest because it shares characteristics with both the *Bacillus* and *Lactobacillus* genera [16]. Compared to vegetative cells such as lactic acid bacteria (LAB), *Bacillus coagulans* is capable of forming microbial spores, which help it to withstand stressful conditions, such as high temperature, acidity, and salinity [17]. Several studies investigating the inclusion of *Bacillus coagulans* in pasta have demonstrated that pasta possesses a buffering capacity and is feasible for probiotic incorporation [18], revealing its potential for foods that require high-temperature treatment.

With regard to LAB, particularly those from *Lactobacillus* and *Bifidobacterium* genera, they are the most commonly used microorganisms in probiotic foods. Compared to *Bacillus coagulans*, LAB are more sensitive to heat treatment. Generally, foods involving high-temperature treatment are not used as a probiotic carrier due to factors affecting the viability of probiotic bacteria [19]. However, as exemplified by spore-formers like *Bacillus coagulans* discussed earlier, certain probiotics can withstand such processing. This presents a particular challenge for those sensitive strains that cannot survive harsh environments. Therefore, including LAB in foods requiring thermal processing can be challenging. While previous studies have demonstrated the potential of incorporating *Bacillus coagulans* into pasta matrices [20] and explored LAB microencapsulation in noodles [21], a systematic understanding of how processing methods affect different types of probiotics in cereal matrices remains limited. Specifically, direct comparisons between heat-sensitive LAB and spore-forming strains within identical carbohydrate foods, and across common processing techniques like extrusion and sheeting, are lacking in the literature. Unlike prior studies focusing on single probiotic strains or one processing method, this study aims to examine the feasibility of two distinct probiotics, *Lactobacillus rhamnosus* GG (LGG) and *Bacillus coagulans* GBI-30 (BC30) bacteria, in the development of probiotic-fortified pasta and noodles, and to compare their resistance during production and cooking processes when incorporated in pasta and noodles. The bacterial survival of LGG and BC30 during production and cooking, as well as the water absorption and cooking loss of pasta and noodles, were investigated and compared. The goal was to enhance the therapeutic potential of cereals through probiotic fortification and bring health benefits to consumers by improving the intestinal microbial balance.

## 2. Materials and Methods

### 2.1. Bacterial Strains

The strains LGG (*Lactobacillus rhamnosus* GG, ATCC 53103) and BC30 (*Bacillus coagulans* GBI-30, 6086) were selected for use as probiotic ingredients in the pasta- and noodle-making process. LGG was purchased from Culturelle^®^ (Amerifit, Inc, Cromwell, CT, USA) with a declared concentration of 5.0 × 10^9^ CFU/g. BC30 was obtained in a freeze-dried form with a claimed concentration of 1.5 × 10^10^ CFU/g under the name of GanedenBC^30^ from Kerry Inc. (Beloit, Rock County, WI, USA) and stored under cool, dry conditions.

### 2.2. Preparation of Pasta and Noodles

The durum wheat semolina (Bellata gold milling industry, Tamworth, New South Wales, Australia) used had the following composition: ash 0.6%, lipid 1.4%, protein 13%, and starch 73%. Of this, 32% was amylose (dry mass) [22]. A 350 μm sieve was used to remove large particles from the commercial semolina. Tap water was boiled and then cooled to room temperature before use.

Spaghetti pasta was produced by extrusion using a Parallel Twin-Screw Extruder (Process-11, Thermo Fisher Scientific Inc., Waltham, MA, USA) according to the experimental protocol reported by Bin et al. (2023) with modifications [22]. The ratio of the length to the barrel diameter was 40:1, and the diameter of the stainless-steel die hole was 2.0 mm. The feed rate and the water feed rate were set to 0.440 kg/h and 165 mL/h, respectively, resulting in a dough moisture content of 27.28%. Consistent operating conditions with a screw speed of 400 rpm were used for each batch of pasta (200 g). The temperature profile from the feeding zone to the die zone was 25, 30, 35, 40, 45, 50, and 45 °C. Fresh pasta was collected and cut into strands with a length of approximately 22 cm. A climate chamber (Memmert GmbH & Co. KG, Schwabach, Germany) with continuous temperature and humidity monitoring was used to dry the fresh pasta at 27 ± 0.5 °C, 70 ± 2% humidity, for 48 h. The dried pasta with a final moisture content of 13.54% was sealed in a plastic bag and stored in a dry, and cool place.

Noodles were produced by sheeting and cutting (Figure 1), first by mixing 200 g of durum wheat semolina with 80 mL of sterilized tap water to form a dough with a total water content of 40%. The resulting dough was kneaded for 3 min and then covered with aluminum foil cover for 10 min at room temperature to allow hydration. This was followed by a second 3 min kneading to ensure proper consistency; the dough was then rolled into neat sheets with a thickness of approximately 2.0 ± 1.0 mm and cut into 2.0 ± 1.0 mm strips.

Probiotic pasta and noodle samples were produced by adding probiotic powder to the semolina flour to achieve a final concentration of around 10^8^ CFU/g. The extruder was sterilized by heating up to 100 °C for 10 min and cooling down before probiotic runs. The pasta and noodles without the addition of probiotic powder were used as controls. The control samples were processed first prior to probiotic batches.

### 2.3. Determination of Water Absorption and Cooking Loss

The cooking behaviour of the fresh pasta and noodle samples was determined. Eight grams of pasta or noodle samples were cooked in 250 mL of boiling (100 °C) distilled water for 7 min. The optimal cooking time was attained when the white core in the centre of the pasta or noodle disappeared (AACC Method 66-50.01). The cooked pasta and noodle samples were drained for 1 min and weighed to measure their final weight. Water absorption was determined using the equation given below:(1)$$\%\;Water\;absorption\;=\;\left(\frac{cooked\;pasta\;weight}{uncooked\;pasta\;weight}-1\right)\times 100$$

Cooking loss was identified based on the loss of solid content in pasta or noodles (AACC Method 66-50.01) during cooking. The cooking water was transferred to a pre-weighed beaker and then dried in an oven at 105 °C for 24 h. The cooking loss was determined using the equation given below:(2)$$\%\;cooking\;loss\;=\;\frac{residue\;weight}{sample\;weight}\times 100$$

### 2.4. Microbiological Analysis

The viable cell counts of LGG and BC30 were determined in various sample types, including fresh samples (flour mixture, dough, fresh pasta, and fresh noodles), dried noodles, cooked probiotic noodles, and cooking water. During pasta and noodle production, samples were collected at key processing stages. For pasta, samples were taken from the flour mixture and fresh pasta (post-extrusion). For noodles, samples were collected from the flour mixture, fresh dough, and freshly cut noodles. To evaluate the homogeneity of probiotic distribution, triplicate samples were taken from three distinct locations within the blended flour (top, middle, and bottom).

For both strains, 0.5% (*w*/*v*) sterile peptone solution was used as the diluent. The enumeration of LGG was carried out by the standard plate count method for microbiological analysis. One gram of each sample was weighed and suspended in 9 mL of a sterile peptone solution using a spatula. The suspensions were then serially diluted to 10-fold dilutions, followed by spreading 0.1 mL of the appropriate dilution on a de Man Rogosa Sharpe (MRS) agar plate for a 48-h stationary incubation at 37 °C. Plates containing 25–250 colonies were counted to calculate the number of viable cells in each sample. For the enumeration of BC30, the appropriate decimal dilutions were heat-treated in a water bath at 75 °C for 30 min and spread-plated on a glucose yeast extract agar, followed by incubation at 40 °C for 48 h. Typical BC30 colonies were counted after incubation. Viability tests were carried out in duplicate, with each measurement performed in triplicate to obtain an average value.

### 2.5. Statistical Analysis

All results are expressed as the mean ± standard deviation (SD). Duplicates of each experimental condition were included to ensure statistical robustness. Statistical analysis was performed using one-way ANOVA and Tukey’s Honestly Significant Difference (HSD) test using SPSS version 3.0 (SPSS Inc., Chicago, IL, USA). Differences were considered significant at *p* < 0.05.

## 3. Results

### 3.1. Impact of Pasta and Noodle Production on Bacterial Survival

During pasta and noodle production, samples were collected from the flour mixture and fresh pasta for extruded pasta, and from the flour mixture, fresh dough and fresh cut noodles for noodles. The viable cell counts from different locations within the flour mixtures (top, middle, and bottom) showed minimal variation, as reflected by the low standard deviations (Appendix A), confirming a homogeneous distribution of probiotics prior to processing. The viability dynamics of LGG and BC30 were systematically monitored across production stages (Figure 2). As shown in Figure 2, the viable cell counts of both LGG and BC30 decreased during processing. Extrusion processing imposed substantial stress on both strains, with LGG exhibiting higher sensitivity (Figure 2a). Initial inoculum level (9.14 ± 0.09 log CFU/g for LGG; 9.20 ± 0.07 log CFU/g for BC30) declined post-extrusion to 7.50 ± 0.04 log CFU/g (2.29% survival) and 8.09 ± 0.04 log CFU/g (7.76% survival), respectively.

In contrast, sheeting-based noodle production better preserved probiotic viability (*p* < 0.05) (Figure 2b). Post-mixing reductions were limited to 0.21 log CFU/g for LGG (from 9.60 ± 0.05 to 9.39 ± 0.03 log CFU/g) and 1.05 log CFU/g for BC30 (from 9.69 ± 0.03 to 8.64 ± 0.03 log CFU/g). Subsequent kneading and cutting further reduced counts to 8.92 ± 0.07 and 8.42 log ± 0.02 CFU/g for LGG and BC30, respectively. Compared to extrusion, a significantly higher bacterial viability was obtained from samples made by sheeting and cutting (*p* < 0.05). Interestingly, more LGG survived during sheeting and cutting, whereas BC30 showed a higher viability in pasta produced by extrusion, suggesting a strain-specific processing tolerance.

### 3.2. Survival of LGG and BC30 During Noodle Production and Cooking

Noodle samples were selected to investigate the impact of drying and cooking on the viability of LGG and BC30. Microbiological analyses were conducted on dried probiotic noodles, uncooked (0 min boiling) and cooked (7 min boiling) fresh noodles, as well as the cooking water. According to Figure 3, drying induced significant viability losses for both LGG and BC30 (*p* < 0.05). Dehydration reduced the viable cell count of LGG by 2.43 log CFU/g (from 8.92 ± 0.07 to 6.49 ± 0.05 log CFU/g) versus a decline of 0.93 log CFU/g for BC30 (from 8.42 log ± 0.02 to 6.49 ± 0.05 log CFU/g CFU/g). Comparing the two, LGG experienced a significantly greater drop in viability during drying than BC30. Furthermore, the cooking process caused a significant reduction in the viable cell count for both strains (*p* < 0.05), especially BC30. Uncooked samples contained 7.35 ± 0.06 log CFU/g LGG and 7.23 ± 0.02 log CFU/g BC30, which then decreased to 6.88 ± 0.06 and 5.75 ± 0.02 log CFU/g, respectively, after cooking. In addition, 4.38 log CFU/mL of LGG was detected in the cooking water after 7 min of cooking, whereas BC30 cells did not leach.

### 3.3. Cooking Behaviours of Noodles

Table 1 illustrates the water absorption and cooking loss values of pasta and noodle samples after 7 min of cooking in boiling water. According to the table, cooking behaviour analysis revealed distinct matrix characteristics. While water absorption showed no significant difference between the pasta and noodles (*p* > 0.05), the addition of probiotic powders significantly decreased the water absorption (*p* < 0.05). As expected, dried noodles tended to absorb more water than fresh samples (*p* < 0.05).

In terms of the cooking loss, a significant difference was observed between the pasta and noodles (*p* < 0.05). Compared to the noodle samples (ranging from 2.40 ± 0.24% to 4.04 ± 0.49%), more residues were left in the cooking water during the cooking of extruded pasta (ranging from 10.60 ± 0.09% to 19.05 ± 0.20%) (Table 1). Moreover, probiotic fortification reduced cooking loss. Drying noodles from fresh samples did not significantly affect this parameter in the subsequent boiling step (*p* > 0.05).

## 4. Discussion

The distinct survival patterns of LGG and BC30 across processing stages reveal critical insights into probiotic viability optimization. The viability loss of LGG and BC30 during extrusion correlates with cumulative thermal and mechanical stress. During extrusion, a high screw speed at 400 rpm induced temperature elevation at the cooling to about 55 °C, which may cause bacterial inactivation. The greater viability loss of LGG during extrusion aligns with the established thermo-sensitivity profiles of *Lactobacillus* species [23]. In terms of BC30, it demonstrated a greater resistance to extrusion, likely due to its inherent spore-forming capacity [17], as evidenced by the smaller viability gap between extruded pasta and sheeted noodles. Notably, despite substantial losses, both strains retained post-extrusion viability exceeding the minimum therapeutic threshold of 10^6^–10^7^ CFU/g [12], underscoring the feasibility of extrusion for probiotic cereal products. Compared to extrusion, sheeting and cutting preserved the bacterial viability more effectively, probably attributable to the milder processing conditions.

Drying and cooking further influenced their viability. In comparison, LGG demonstrated a greater viability loss during drying than BC30 (2.43 vs. 0.93 log CFU/g). The fundamental differences in their cell physiology can be the explanation. As a vegetative bacterium, LGG lacks specialized structures to withstand dehydration stress. Drying may disrupt membrane integrity, denature proteins, and damage DNA in metabolically active cells [23]. Conversely, BC30 is a spore-forming probiotic. Its spores possess a protective structure and exhibit metabolic dormancy, enabling exceptional resistance to desiccation [24]. The core dehydration mechanisms in spores, including the accumulation of dipicolinic acid and small acid-soluble proteins, further minimize macromolecular damage during water loss [25]. This inherent resilience allows BC30 spores to maintain viability under drying conditions that critically impair vegetative cells like LGG. Interestingly, although BC30 survived the drying step better than LGG, it experienced a more substantial loss in viability than LGG during the 7 min cooking process. This apparent contradiction may be attributed to the physiological states of the bacteria during cooking. BC30 spores, while inherently heat-stable, likely germinated upon hydration and exposure to the nutrient-rich noodle matrix during boiling, converting it into metabolically active vegetative cells [24,26]. Vegetative cells of *Bacillus* spp. are significantly more heat-sensitive than their dormant spores [25]. In contrast, a non-spore-forming LAB such as LGG may have developed partial stress adaptation during prior processing stages, enhancing its transient tolerance to sub-lethal thermal stress [27]. This mechanistic hypothesis aligns with previous work reporting similar viability drops in *Bacillus* during short-term thermal processing. For example, Kalkan et al. (2020) [28] prepared Turkish noodles supplemented with microencapsulated *Bacillus clausii* and vegetables. Unfortunately, there was a drop in the viable cell count throughout the processing step, reaching 5.02 to 5.10 log CFU/g after cooking. Although the noodle dough contained a significant number of probiotic microorganisms, the viability of *Bacillus clausii* in cooked noodles was lower than the therapeutic recommended level of probiotics [28]. A similar outcome was obtained by Fares et al. (2015), who developed a functional pasta containing BC30 with added barley β-glucan [20]. A reduction in bacterial counts was observed during the cooking process, and the level of reduction depended on the cooking time. As expected, the longer the cooking time, the greater the loss of viable cells [29]. To stimulate the cooking process and shorten the cooking time, they suggest adding NaCl to the water during cooking since it did not negatively impact the viability of probiotics [20]. Therefore, probiotic-fortified pasta should be produced with an appropriate thickness to avoid a prolonged cooking time and to guarantee the survival of the bacterial strain.

Moreover, the viability of LGG probiotic cells was higher in the fresh cooked noodles than in the dried noodles, which aligns with a previous study by Rajam et al. (2015) [21]. They used microencapsulated *Lactobacillus plantarum* MTCC 5422 in probiotic noodle production and obtained a 93.63% cell survival rate. The fresh samples were further dried by room-temperature drying at 28 °C or high-temperature drying at 55 °C, reducing the survival rate to 80.29% and 64.74%, respectively. After cooking, the fresh sample had a survival rate of 62.42%, whereas the dried noodles experienced a complete loss of cell viability, suggesting that fresh pasta might serve as a more suitable carrier system for delivering viable probiotic cells [21]. Thus, compared to pasta production, which involves milder mechanical stresses, drying and high-temperature heat treatment (e.g., boiling) had a more detrimental impact on bacterial viability.

There is a general consensus that cooking performance is associated with the interactions between water and macromolecules such as starch and protein [22]. The content of starch and protein plays an important role in determining the cooking behaviour of pasta and noodles [30]. The simultaneous starch gelatinization and protein coagulation which occurred during cooking are responsible for major structural changes that affected the cooking quality of the pasta and noodles [31]. In the current study, probiotic-fortified pasta or noodles retained a lower water content than the control samples, suggesting a lower moisture uptake due to their reduced water-holding capacity [21]. Our results are consistent with those reported by Konuray and Erginkaya (2020) [32], who found that pasta fortified with probiotics absorbed less water during cooking. Moreover, an increase in water absorbed by dried noodles suggests easier water penetration, probably due to the gluten weakening during the drying process [33], which allowed for a more rapid water diffusion into the noodles and thus faster starch swelling [31].

In addition, solid loss is an important factor when determining cooking quality [34]. A higher cooking loss in extrusion implies a greater extent of macromolecule solubilization. Moreover, the loss of solids in the cooking water relates to structure, suggesting a more open or disrupted matrix in extruded pasta [35]. By comparison, noodle samples demonstrated a lower cooking loss than pasta, reflecting the formation of a stronger network that prevented starch granules from escaping during cooking. Thus, the sheeting and cutting approach may be preferred for producing probiotic noodles since cooking loss is recommended to be controlled within 6.5–7.6% for wheat flour noodles [36]. Interestingly, a reduction in the cooking loss and solid leaching into cooking water observed in probiotic-fortified samples indicates less disruption of the gluten network and fewer breaks in the protein matrix [21], which supports the feasibility of probiotic-enriched noodles.

## 5. Conclusions

This study developed a type of probiotic-fortified pasta and noodles with the addition of LGG or BC30 via extrusion or traditional kneading, rolling and cutting. Microbiological analysis showed a final concentration of at least 8 log CFU/g in fresh noodles made by cutting and sheeting, and a slightly lower probiotic load in pasta produced by extrusion. LGG and BC30 maintained a relatively high viability after 7 min of cooking in noodles, reaching 6.88 and 5.75 log CFU/g, respectively. Considering the requirement that probiotic food products should contain a minimum of 10^6^ to 10^7^ CFU/g viable bacteria for optimum functionality, the LGG count exceeds the minimum recommended level for probiotic efficacy, while BC30 falls marginally below this threshold. Coupled with cooking loss within the recommended level, the fresh probiotic noodle in this study may qualify as a promising candidate for probiotic functional foods, pending sensory evaluation, shelf-life stability assessments, and in vivo efficacy studies to confirm clinical benefits. Due to the growing consumer demand for non-dairy and heat-treated functional food, novel cereal-based products such as probiotic-fortified noodles may offer a new alternative for probiotic supplementation.

## Figures and Tables

**Figure 1 foods-14-02250-f001:**
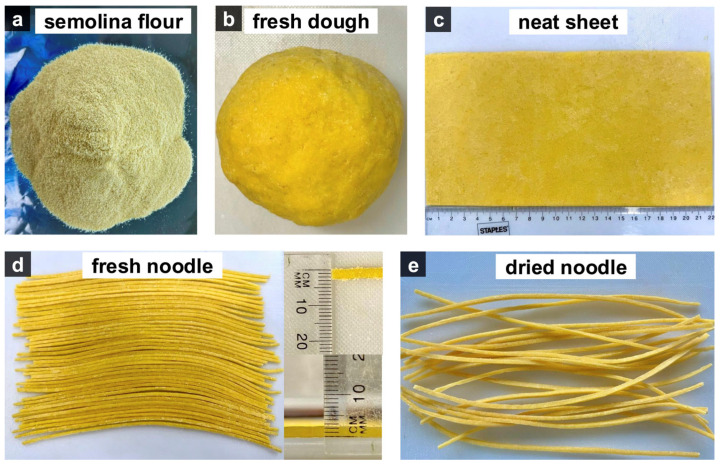
Images of noodle production from semolina flour (**a**), fresh dough (**b**), neat sheet (**c**), fresh noodle (**d**), and dried noodle (**e**).

**Figure 2 foods-14-02250-f002:**
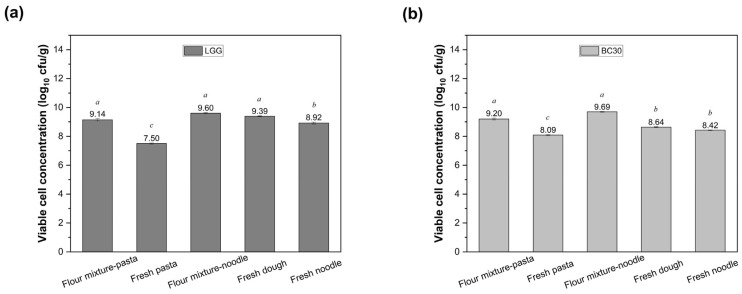
Viability of LGG (**a**) and BC30 (**b**) (log CFU/g) during the production of probiotic-fortified pasta and noodles. Values with different letters (for either pasta or noodle samples) are significantly different (*p* < 0.05).

**Figure 3 foods-14-02250-f003:**
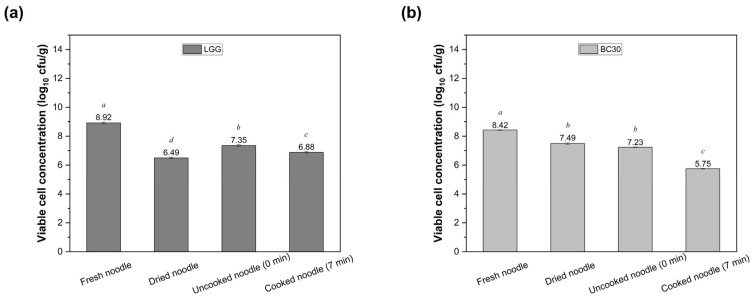
Viability of LGG (**a**) and BC30 (**b**) (log CFU/g) during the drying and cooking process of probiotic-fortified noodle samples made by cutting. Values with different letters are significantly different (*p* < 0.05).

**Table 1 foods-14-02250-t001:** Water absorption (%) and cooking loss (%) of pasta and noodle samples with or without (control sample) the addition of LGG and BC30 ^1,2^.

Sample	Water Absorption (%)	Cooking Loss (%)
Pasta		
Fresh		
Control Pasta	64.36 ± 2.15 ^c^	18.79 ± 0.63 ^a^
LGG Pasta	59.38 ± 0.38 ^d^	10.60 ± 0.09 ^e^
BC30 Pasta	47.31 ± 0.24 ^e^	17.24 ± 0.11 ^b^
Dried		
Control Pasta	91.15 ± 7.04 ^a^	19.05 ± 0.20 ^a^
LGG Pasta	83.80 ± 0.35 ^b^	13.99 ± 0.32 ^d^
BC30 Pasta	60.07 ± 0.38 ^d^	15.84 ± 0.59 ^c^
Noodle		
Fresh		
Control Noodle	65.94 ± 0.28 ^b^	4.04 ± 0.49 ^a^
LGG Noodle	55.82 ± 2.27 ^c,d^	2.98 ± 0.16 ^d^
BC30 Noodle	52.05 ± 2.81 ^d^	3.67 ± 0.26 ^a,b^
Dried		
Control Noodle	69.44 ± 0.32 ^a^	3.45 ± 0.07 ^b^
LGG Noodle	60.97 ± 4.60 ^b,c^	2.40 ± 0.24 ^e^
BC30 Noodle	56.50 ± 2.11 ^c,d^	3.03 ± 0.13 ^c,d^

^1^ Data are presented as mean ± standard deviation based on triplicate tests. ^2^ Values with different letters within the same column (for either pasta or noodle samples) are significantly different (*p* < 0.05), as determined by one-way ANOVA followed by Tukey’s HSD test.

## Data Availability

The original contributions presented in the study are included in the article, further inquiries can be directed to the corresponding authors.

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
