# Peer review of "Fortification of Cereal-Based Food with Lactobacillus rhamnosus GG and Bacillus coagulans GBI-30 and Their Survival During Processing"

_foods, 2025, doi:10.3390/foods14132250_

Round 1
Reviewer 1 Report
Comments and Suggestions for Authors
This manuscript addresses the incorporation of L. rhamnosus GG (LGG) and B. coagulans GBI-30 (BC30) into pasta and noodles, focusing on their survival during food processing and cooking. The work is relevant and addresses the demand for non-dairy probiotic foods. The experimental design is generally sound, the statistical analysis is appropriate, and the results are clearly presented. However, key details are missing in the methods, some interpretations are not sufficiently supported, and certain conclusions should be more cautiously worded. Some figures and tables require better annotation and units.
Line 36–38: Please add more precise history or reference about the origin of the term functional food
Line 48: Please add references of specific studies on cereal fibre as a prebiotic.
Line 54–56: "A minimum of 10^6 to 10^7 CFU/g... is recommended in probiotic food products... Please add a citation about current, authoritative guidelines (e.g., ISAPP, FAO/WHO) rather than secondary sources.
Line 72–75: "Generally, foods involving high-temperature treatment are not used as a probiotic carrier..." Please consider mentioning successful examples such as spore-formers for exception
Line 79–85: Please consider clearlying state here your study’s novelty in comparison to previous work.
Line 89: Please mention the exact strain number used (ATCC or DSM number) and batch/lot for reproducibility.
Line 90–91: “BC30 was obtained in a freeze-dried form” Please Indicate any storage conditions, rehydration process, and how or if the cell counts were confirmed pre-use.
Line 94–95: “Durum wheat semolina...ash 0.6%, lipid 1.4%, protein 13%, starch 73%...” any method used here for composition analysis and batch/lot.?
Line 98–107: Please Include die material, possible cross-contamination controls, and reference to preliminary optimization of extrusion conditions.
Line 108–110: please clarify whether humidity and temperature were monitored throughout and whether pasta was turned/flipped during drying
[Correction] Line 117: “rolled into neat sheets with a thickness of approximately 2 mm and cut into 2-mm strips.” how did you measure the thickness/width ?
Line 122–123: “to achieve a final concentration around 10^8 CFU/g.” did verify how homogeneious the probiotic distribution in the dough/flour verified?
[Correction] Line 126–127: “Eight grams of pasta or noodle samples were cooked in 250 mL of boiling...water for 7 min.” Please provide some rationale for the sample-to-water ratio, and whether this is standardized in the literature.
Line 128: “when the white core...disappeared (AACC Method 66-50.01)” Please mention if endpoint was checked by a single operator or multiple, and whether blind to treatment.
Line 142–145: Please specify diluent for each strain and any selective agents used to differentiate LGG/BC30 from background.
Line 148: “heat-treated in a water bath at 75°C for 30 min...” why this time-temperatue domain? does it kill all non-spore-formers? Was this verified?
Line 165–166: “Initial inoculum level...declined post-extrusion to 7.50 ± 0.04 and 8.09 ± 0.04 log CFU/g, respectively.” Please provide percent survival as well for context; include reference to regulatory/industry targets.
Line 225–226: “BC30 experienced a more substantial loss in viability than LGG…” Please discuss and hypothesize the mechanistic differences for this result.
Line 240–241: “pasta should be produced with an appropriate thickness…” please If you have data on pasta thickness or cross-section, report it.
Line 246–249: Pleasee consider discussing why drying affected LGG more than BC30 in terms of cell physiology.
Line 283: “LGG and BC30 maintained a relatively high viability after 7-min of cooking in noodles, reaching 6.88 and 5.75 log CFU/g, respectively.” State explicitly whether these levels meet international guidelines for probiotic efficacy post-processing.
Line 287: “the fresh probiotic noodle in this study qualifies as an acceptable probiotic functional food.” This is an overstament, please state “may qualify, pending sensory, shelf life, and in vivo efficacy studies.”
Author Response
Pls see attachments

Reviewer 2 Report
Comments and Suggestions for Authors
The presented manuscript shows rather simple, but sensible and straightforward experiment with interesting results.
- 12 ‘evolving beyond satiety’?
- 53 I don’t think that ‘vehicle’ is proper word.
- 59 Embedded.
- 88-92 Add the species of the microorganism also in the Materials and Section, it shouldn’t be only given in Introduction.
- 105 To what temperature?
- 122-124 What are the ingredients of the ‘probiotic powder’, other than microorganisms? This can be important. And how much of the ‘probiotic powder’ was added? The amount of
- 153-155 Standard deviation. What software was used?
Figure 2 – homogenous groups aren’t shown?
Table 1 something is wrong with the homogenous groups. You state, that the ‘same letter in the same column are significantly different’? It doesn’t make much sense. Also, some data do not have letters. Also, ‘in the same column’? So, do you compare dried control pasta to fresh noodles? Also doesn’t make much sense.
Figure 3 – once again, the homogeneity of the samples is really unclear, it is not obvious what is compared to what.
- 219 Thermo-sensitivity.
Author Response
Pls see attachments
